# GRR-CoCa: Leveraging LLM Mechanisms in Multimodal Model Architectures

## Abstract

State-of-the-art (SOTA) image and text generation models are multimodal models that have many similarities to large language models (LLMs). Despite achieving strong performances, leading foundational multimodal model architectures frequently lag behind the architectural sophistication of contemporary LLMs. We propose GRR-CoCa, an improved SOTA Contrastive Captioner (CoCa) model that incorporates Gaussian error gated linear units, root mean squared normalization, and rotary positional embedding into the textual decoders and the vision transformer (ViT) encoder. Each architectural modification has been shown to improve model performance in LLMs, but has yet to be adopted in CoCa. We benchmarked GRR-CoCa against Baseline CoCa, a model with the same modified textual decoders but with CoCa's original ViT encoder. We used standard pretraining and fine-tuning workflows to benchmark the models on contrastive and generative tasks. Our GRR-CoCa significantly outperformed Baseline CoCa on the pretraining dataset and three diverse fine-tuning datasets. Pretraining improvements were 27.25% in contrastive loss, 3.71% in perplexity, and 7.15% in CoCa loss. The average fine-tuning improvements were 13.66% in contrastive loss, 5.18% in perplexity, and 5.55% in CoCa loss. We show that GRR-CoCa's modified architecture improves performance and generalization across vision-language domains.

## 1 Introduction

Transformers have resulted in countless advancements in deep learning across various domains (Vaswani et al., 2017), especially natural language processing (NLP) (Vaswani et al., 2017; Kenton & Toutanova, 2019; Dubey et al., 2024). Causal large language models (LLMs), including Meta's Llama series, have especially improved. These innovations have enabled modern chatbots to perform complex tasks such as code completion, detailed data analysis, mathematical problem solving, and general question answering (Dubey et al., 2024; Liu et al., 2024; Achiam et al., 2023).

The transformer architecture has been applied to other modalities such as computer vision and audio (Alexey, 2020; Radford et al., 2023). State-of-the-art (SOTA) performance has been achieved using transformers as the backbone architecture in many computer vision tasks (Alexey, 2020; Radford et al., 2023). Multimodal models are an extension of transformers that can be prompted by multiple information streams, e.g., text and images (Dubey et al., 2024). These models can effectively replicate human reasoning and produce outputs that surpass those generated in previous work (Yu et al., 2022). Multimodality models further build upon the unimodal text decoder or unimodal visual encoder transformers (Dubey et al., 2024). However, multimodal models result in complex spatial, textual, and relational interdependencies, which produce a difficult task for even transformer-based deep learning models (Yu et al., 2022). Producing a machine learning model that can capture these complex dependencies at a deep level is paramount for creating high-performing and useful models.

The Contrastive Captioners (CoCa) model proposed by Yu et al. (2022) in 2022 is a current SOTA model with the fine-tuned CoCa visual encoder achieving the SOTA performance on top 1% accuracy on ImageNet. While CoCa introduced transformer-based approaches that were novel at the time in both its visual encoder and textual decoders, many modern state-of-the-art models, such as Meta's Llama series, have since

incorporated newer architectural improvements (Dubey et al., 2024; Yu et al., 2022). Recent literature has discovered that incorporating modern LLM architectures has the ability to increase performance in both unimodal language and visual transformer models (Dubey et al., 2024; Chu et al., 2024). There is a critical need for the older SOTA CoCa model to be updated.

To address this challenge, we modified the CoCa model's textual decoder layers with Gaussian error gated linear units (GEGLUs) in the feedforward layer, the replacement of layer normalization (LayerNorm) with root mean squared normalization (RMSNorm) in the pre-normalization location, and the replacement of the absolute positional encoding with rotary positional embedding (RoPe) (Jiang et al., 2024; Shazeer, 2020; Su et al., 2024). This implementation aligns image-captioning models with current SOTA unimodal visual and LLM models, resulting in improved performance of the multimodal models (Dubey et al., 2024; Chu et al., 2024; Team et al., 2024; Adler et al., 2024; Jeevan & Sethi, 2022).

Countless modern SOTA vision transformers (ViTs) also do not use modern LLM architectures. Previous work suggests that using RoPe instead of absolute positional encoding produces a higher-performing ViT (Jeevan & Sethi, 2022; Chu et al., 2024). Other promising architectures for ViTs are GEGLUs and RMSNorms, as they have been shown to improve performance on NLP and visual tasks (Shazeer, 2020; Zhang & Sennrich, 2019; Chu et al., 2024). Based on this, we propose GRR-CoCa, an updated CoCa model with LLM architectures within the ViT of the CoCa model to improve the model's ability to produce more feature-rich image latent representations, which can then be leveraged by downstream models.

To summarize:

- We produced a Baseline CoCa model as a control by modifying its textual decoders to include GEGLUs, RMSNorm, and RoPE. This aligned it with modern SOTA LLM transformer architectures (Dubey et al., 2024).

- We introduced GRR-CoCa. The model retained the same enhanced textual decoders as Baseline CoCa, while also incorporating a visual encoder equipped with GEGLUs, RMSNorm, and RoPE. This ensured architectural consistency across modalities, leading to increased performance.

- We trained the GRR-CoCa and Baseline CoCa models in a standard pretraining to fine-tuning workflow. We studied how GRR-CoCa's architecture increased the model's capacity to fit on both the pretraining dataset and three diverse fine-tuning tasks.

- We quantified the significant improvements in performance that can be gained in multimodal transformer models by architectural modifications with almost no increase in model parameter size.

## 2 Related Work

### 2.1 Modern Improvements in LLM Transformers

The architectures of the previously mentioned open-source SOTA LLM models differ from the original decoder-only model proposed in "Attention is all you need" (Vaswani et al., 2017; Yang et al., 2024; Adler et al., 2024; Dubey et al., 2024; Team et al., 2024). The original architecture employs absolute positional encoding, LayerNorms, a post-norm transformer architecture, and a one-hidden-layer feedforward network (Vaswani et al., 2017). Newer architectures include iterations on the original decoder-only transformer, targeting improvements in efficiency and expressiveness (Dubey et al., 2024; Vaswani et al., 2017). In contrast, Llama 3.1 uses RoPe, RMSNorm, a pre-norm transformer architecture, and a gated linear unit (GLU) in the feedforward network (Dubey et al., 2024). Modifications such as RMSNorm and pre-norm architecture aim to improve model efficiency, while GLU and RoPe enhance model expressiveness (Zhang & Sennrich, 2019; Jiang et al., 2024; Shazeer, 2020; Su et al., 2024).

#### 2.1.1 Gaussian Error Gated Linear Units (GEGLUs)

The GLU ViT is a transformer architecture designed to enhance model expressiveness by modifying the feedforward layer (Shazeer, 2020). In a standard ViT, the feedforward layer consists of an input layer, a

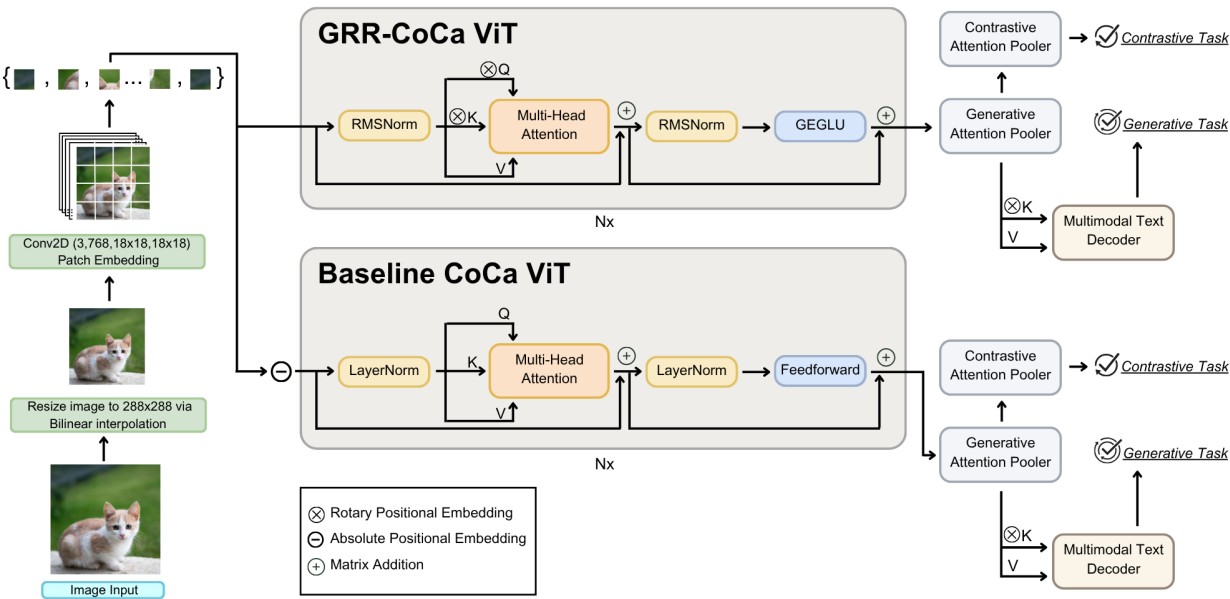

Figure 1: Overview of GRR-CoCa vs. Baseline-CoCa ViT-based visual encoder architecture. The GRR-CoCa model uses GEGLUs, RMSNorms, and RoPe. The Baseline CoCa model uses feedforward layers, LayerNorms, and absolute positional encoding.

hidden layer, and an output layer, with the Gaussian error linear unit's (GELU) activation function being applied to the hidden layer (Alexey, 2020). The implementation of a GLU modifies the first half of the feedforward structure. Instead of a single linear transformation followed by bias and activation functions, the input is passed through two parallel linear transformations with accompanying bias (Shazeer, 2020). Then, one of the linear transformations is passed through an activation function (such as GELU) (Shazeer, 2020). Afterwards, the Hadamard product is taken between the output from the activation and the output from the other linear transformation (Shazeer, 2020). This produces a gated output that is passed to a final linear transformation to produce the final output of the layer (Shazeer, 2020). The following equation demonstrates this modification to the feedforward as

$$\text{GLU} = (\sigma(xW + Wb) \odot (xV + Vb))O + Ob \tag{1}$$

where

- $x$ is the input sample,

- $W$, $V$, and $O$ are the weights of the linear transformations,

- $Wb$, $Vb$, and $Ob$ are the accompanying weights for the output of each linear transformation, and

- $\sigma$ is any arbitrary activation function (Shazeer, 2020).

Utilizing these modified feedforward network architectures has produced better perplexity scores on causal language modeling tasks and improved performance on downstream NLP tasks such as semantic classification (Shazeer, 2020). The consistent increase in performance it yields in many NLP applications and datasets provides strong support for the inclusion of these in vision transformers as well.

### 2.1.2 Root Mean Squared Error Normalization (RMSNorm)

LayerNorm was found to be imperative to stabilize any transformer architecture (Vaswani et al., 2017). However, it does lead to computational overhead when handling deep neural networks due to its trainable

parameters for recentering and rescaling invariance of the input vector (Zhang & Sennrich, 2019). The simpler alternative to LayerNorm, proposed in 2019 by Zhang & Sennrich (2019), was the RMSNorm. The RMSNorm modifies the equation to remove the recentering parameter and eliminates the need to compute the mean statistic, resulting in reduced computational demand for this layer. RMSNorm has also been shown to improve the evaluation metrics of certain tasks over the standard LayerNorm. Both the increased efficiency and frequent superior performance support our hypothesis that including RMSNorm in GRR-CoCa will increase its performance on generative captioning tasks.

### 2.1.3  Rotary Positional Embedding (RoPe)

The absolute positional encoding used by Vaswani et al. (2017) is typically added to the token embeddings before being passed to the transformer blocks. An improvement over this original method, RoPe, was proposed by Su et al. (2024) in 2021. This method eliminates the original absolute positional encoding, instead applying a set non-trainable linear transformation that effectively rotates each query and key embedding a set distance, where the total angle it rotates is dependent on the position index the embedding is in the sequence. Employing RoPe resulted in an improvement in performance metrics in multiple NLP tasks, such as machine translation. RoPe can be directly applied in ViT's image patch embeddings and has been shown to improve performance when incorporated in a ViT performing image classification tasks (Jeevan & Sethi, 2022). Incorporating RoPE into the visual encoder of a multimodal model is an underexplored idea with significant potential.

## 2.2  Vision Transformer (ViT)

The specific ViT architecture that will be modified by the previously discussed LLM architectural modification was proposed by Alexey (2020) in 2020. The ViT utilized the traditional transformation from the original transformer, except instead of using a tokenized input sequence with a trainable embedding layer, it broke the input images into "patches" using either linear formation or convolutions. A convolutional operation with a specified window size and stride is used to divide the image into patches and simultaneously project the input channels (typically 3 for RGB or 1 for grayscale) into the embedding dimension of each token. At the time of publication of "An Image is Worth 16x16 Words" (Alexey, 2020), the ViT produced SOTA performance at many different visual tasks and notably resulted in the best accuracy achieved on the ImageNet image classification challenge task.

## 2.3  Mutlimodal Transformers

### 2.3.1  CLIP and ALIGN

Two notable modern multimodal models are CLIP (Radford et al., 2021) and ALIGN (Jia et al., 2021). Both trained large models (a visual transformer for CLIP and an EfficientNet CNN for ALIGN; textual transformer encoders were used in both models) on large datasets of visual and textual data in order to produce the latent representations of an image and its paired text to be more similar to each other. The contrastive loss task performed was to maximize similarity between the image-caption pairs, and maximize dissimilarity between the non-corresponding image-caption pairs in the batch. This self-supervised learning, where the model learns to associate specific images with their textual representation and vice versa, resulted in more robust performance in multimodal tasks but also in unimodality downstream NLP or computer vision tasks as well. The high-level abstraction that a model can learn using both textual and visual information proved to be a robust way for the model to learn.

### 2.3.2  Contrastive Captioners (CoCa)

CLIP and ALIGN exhibited how contrastive learning within multimodal text and image models provides SOTA performance on multiple benchmarks (Radford et al., 2019; Jia et al., 2021). In 2022, Yu et al. (2024) showed the progression of these contrastive training models with their CoCa model (Yu et al., 2022). CoCa utilized transformer architectures within all modalities (Yu et al., 2022). It also built on the simple contrastive loss objectives of CLIP and ALIGN by incorporating another textual decoder that takes the

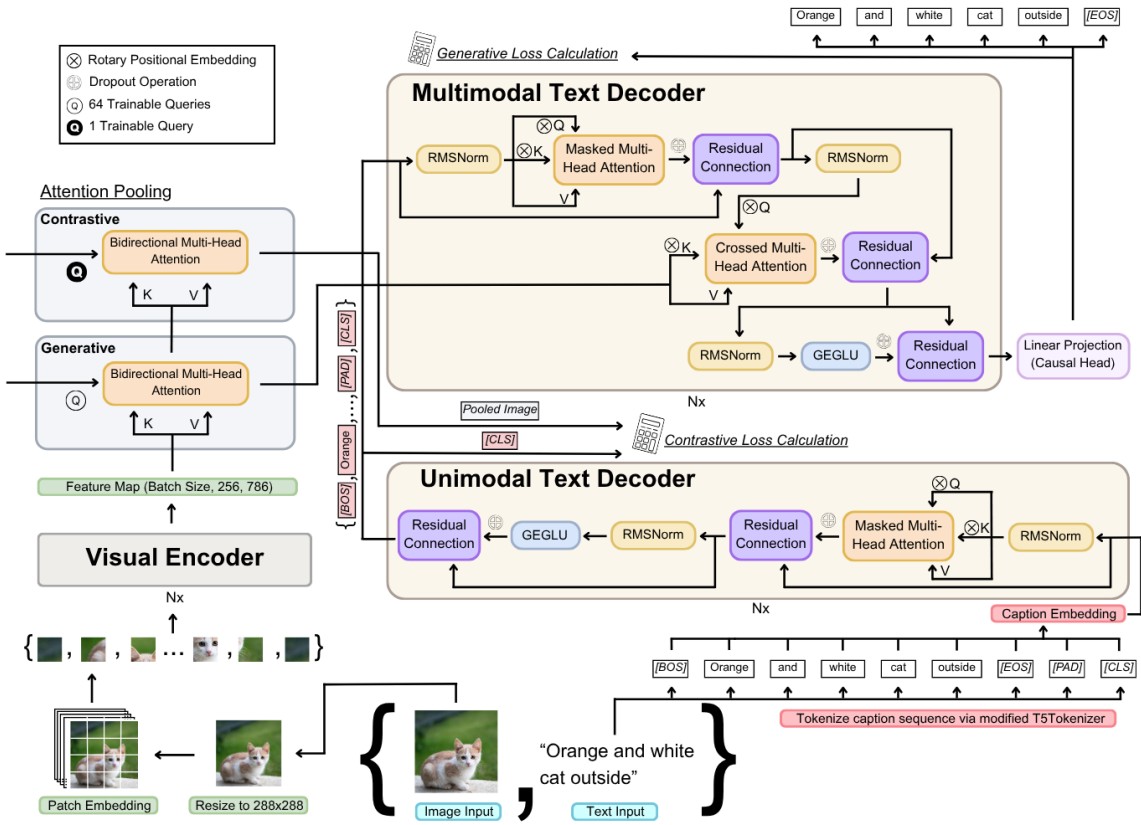

Figure 2: Architectural overview of GRR-CoCa and Baseline CoCa's attention poolers, unimodal textual decoder, and multimodal textual decoder. The decoders utilize GEGLUs, RMSNorms, and RoPe. GRR-CoCa and Baseline CoCa use different visual encoders outlined in Figure 1.

output from the text and image models (Yu et al., 2022). In addition, a caption loss function (simple causal language modeling) is performed by a multimodal textual decoder that takes the output embeddings from the unimodal decoder and image feature map from the visual encoder to perform cross-attention (Yu et al., 2022).

This dual task pretraining allows a model to leverage contrastive self-supervised learning with a generative textual model trained to produce text as output (Yu et al., 2022). This effectively produces a model with a unified understanding where each aspect (visual encoder, unimodal text decoder, or multimodal text decoder) can be used individually or combined on multimodal tasks, NLP tasks, or visual tasks (Yu et al., 2022). The dataset originally used to train CoCa is Google's proprietary JFT-300 M dataset (Yu et al., 2022). This dataset contains around 300 million noisy image-caption pairs (Yu et al., 2022). The results produced SOTA performance, notably using a fine-tuned CoCa visual encoder on the ImageNet image classification dataset (Yu et al., 2022). Currently, this model is still SOTA for image-captioning, but the architecture of both the textual decoder and image encoder is becoming outdated.

## 3 Methods and Experiments

### 3.1 Datasets

#### 3.1.1 Conceptual Captions 12 Million (CC12M)

The pretraining dataset used was the Conceptual Captions 12 million (CC12M) dataset (Changpinyo et al., 2021a; Hugging Face, n.d.). CC12M was used to simulate the larger proprietary JFT image-caption dataset

used by the original CoCa model (Lin et al., 2014; Yu et al., 2022; Sun et al., 2017; Changpinyo et al., 2021b). CC12M contains general image-caption pairs of objects such as a "Victorian dining room" (Changpinyo et al., 2021a). We randomly partitioned CC12M into 10,445,110 image-caption pairs for the training set and 549,743 image-caption pairs for the validation set (95% training, 5% validation).

### 3.1.2 Microsoft Common Objects in Context (MSCOCO)

The first fine-tuning dataset was the Microsoft Common Objects in Context (MSCOCO) dataset (Lin et al., 2014). MSCOCO's samples are similar to CC12M's distribution as they both contain common objects. MSCOCO was pre-split into training and validation sets by the original authors (Lin et al., 2014). Each training example typically had five different captions; therefore, the total number of image-caption pairs in the training set was 591,753, with the validation set containing 25,014 image-caption pairs.

### 3.1.3 Radiology Objects in COntext (ROCO) Version 2

The second fine-tuning dataset was the Radiology Objects in COntext (ROCO) Version 2 dataset, a more complex medical image-captioning dataset. ROCO was used to simulate a transfer learning scenario as it lies farther from CC12M's distribution (Rückert et al., 2024; Eslami et al., 2023). Pretraining on a general knowledge and fine-tuning on a specific medical dataset has been found to be a practical approach in multimodal models such as CLIP and PubMedClip (Eslami et al., 2023). ROCO was pre-split into training and validation sets, with the training set having 69,866 image-caption pairs and the validation set containing 9,904 image-caption pairs (Rückert et al., 2024).

### 3.1.4 Flickr 30K

Lastly, the GRR-CoCa and Baseline CoCa models were fine-tuned on the Flickr 30K dataset (Young et al., 2014). This common benchmark dataset contained around 30k images with five hand-annotated captions per image (Young et al., 2014). The training-validation split used on the dataset was the Karpathy split with 29,000 training images and 1,000 validation images (Karpathy & Fei-Fei, 2015).

## 3.2 Data Preprocessing

To process each dataset's captions into input IDs, we used a modified T5 base tokenizer (Raffel et al., 2020). We added a "beginning of sequence" token (BOS) and a classification token (CLS) to the vocabulary. The BOS token was then prepended to all captions, and the "end of sequence" (EOS) token was added after the last non-pad token in the caption. The native T5 PAD token was used to pad each caption to a uniform length. The CLS token was appended after the PAD tokens. Finally, a single PAD token was appended after the CLS token to produce the correct form for causal language modeling. This process was performed for all captions across all datasets.

All images were resized to $288 \times 288$ pixels using bilinear interpolation, converted to RGB, and scaled to the $[0, 1]$ interval. Lastly, these values were normalized using the standard ImageNet channel mean and standard deviations (Deng et al., 2009).

## 3.3 Model Architectures

We constructed two different multimodal CoCa model variants. Both CoCa model variants used modern LLM architectures (GEGLUs, RoPe, and RMSNorms) in their textual decoder submodels (i.e., unimodal and multimodal decoders) as seen in Figure 2. This ensured that any improvements in results were due to modifications in the visual encoder. "Baseline CoCa" used a ViT submodel that followed the original architecture from Alexey (2020). GRR-CoCa used a modified ViT that exchanged the feedforward network for GEGLUs, absolute positional encoding for RoPe, and LayerNorms for RMSNorms (Su et al., 2024; Shazeer, 2020; Zhang & Sennrich, 2019). Figure 1 illustrates the visual encoders used by GRR-CoCa and Baseline CoCa.

| Architecture Hyperparameter | Value |
|---|---|
| *Embedding Dimension* | 768 |
| *Number of Attention Heads per Model* | 12 |
| *Number of Blocks per SubModel* | 12 |
| *Non-GEGLU Feedforward Hidden Scaler* | 4 |
| *GEGLU Feedforward Hidden Layer Scaler* | 2.7 |
| *Token Context Length* | 64 |
| *Vocabulary Size* | 32,102 |
| *Dropout During Pretraining* | 0.15 |
| *Dropout During Fine-tuning* | 0.1 |

Table 1: Model architecture and tokenizer hyperparameters used in GRR-CoCa and Baseline CoCa.

The pooling utilized is the same attention poolers described in the original CoCa paper (Yu et al., 2022). These are self-attention layers where the keys and the values are derived from the visual encoder output and $n$ queries (Yu et al., 2022). The parameter $n$ determines the number of queries we are training and is the same number of embeddings that are output from this attention block (Yu et al., 2022). For GRR-CoCa, we first sent the visual encoder output (256 image patch embeddings) to a generative pooler that projected the 256 patches to the model's context length of 64 (Yu et al., 2022). This output was then passed as the context in cross attention (Yu et al., 2022). In parallel, this generative output pooler was passed to another contrastive pooler that projected these 64 embeddings to a single embedding used in the contrastive loss function as seen in Figure 2 (Yu et al., 2022). This cascading pooler structure was found to be the best structure tested by Yu et al. (2022).

### 3.4 Model Hyperparameters

The models' hyperparameters were chosen similarly to those of the original CoCa model (Yu et al., 2022). The embedding dimensions of the visual encoder, the unimodal decoder, and the multimodal decoder were all 768. Each encoder/decoder submodel had 12 transformer blocks. The sub-model's attention layers each had 12 attention heads. A universal dropout of 0.15 was applied during pretraining, and a dropout of 0.1 was applied when fine-tuning. The size of the hidden layer in a non-GEGLU feedforward network was calculated as the embedding dimension multiplied by 4. Since adding the additional linear transformation in GEGLUs leads to a higher number of parameters in the model overall if the same $4\times$ scaling factor is used for the hidden layers size in the feedforward network, we instead scaled the size of the hidden layers in the GEGLUs by a factor of 2.7 to keep the parameter sizes of the models similar (Shazeer, 2020). A summary of the models' hyperparameters is provided in Table 1. GRR-CoCa had 0.17% more trainable parameters (394,548,926) than Baseline CoCa (393,863,270). The context length of the model was set to 64 tokens, and the T5 tokenizer with the added tokens resulted in a vocabulary size of 32,102.

### 3.5 Loss Functions

Baseline CoCa and GRR-CoCa were trained simultaneously on contrastive and generative captioning tasks. The contrastive task involves training the models such that their image latent representations and their textual latent representations have similar embeddings.

The contrastive loss function reduces the distance in the embedding space between image-text pairs while increasing the distance between non-matching pairs (Yu et al., 2022). This effectively pushes the model to relate the image latent representation to the textual latent representations. Lower contrastive loss values indicate stronger alignment, reflecting the impact of architectural improvements. For each batch of image-text pairs, the latent representations are evaluated by the contrastive loss as follows (Yu et al., 2022):

$$\mathcal{L}_{\text{Con}} = -\frac{1}{N} \left[ \sum_{i=1}^{N} \log \left( \frac{\exp(x_i^\top y_i / \sigma)}{\sum_{j=1}^{N} \exp(x_i^\top y_j / \sigma)} \right) \right.$$
$$\left. + \sum_{i=1}^{N} \log \left( \frac{\exp(y_i^\top x_i / \sigma)}{\sum_{j=1}^{N} \exp(y_i^\top x_j / \sigma)} \right) \right] \tag{2}$$

where

- $N \in \mathbb{N}$ is the batch size,

- $d \in \mathbb{N}$ is the dimension of each text/image latent representation,

- $x_i \in \mathbb{R}^d$ is the normalized embedding of the image in the $i$-th pair, where $i \in \{1, 2, \ldots, N\}$,

- $y_j \in \mathbb{R}^d$ is the normalized embedding of the text in the $j$-th pair, where $j \in \{1, 2, \ldots, N\}$, and

- $\sigma \in \mathbb{R}$ is the temperature used to scale the logits.

The generative captioning task is trained using autoregressive causal language modeling, in which the model learns to maximize the probability of each token given all preceding tokens (Yu et al., 2022). The formula for this loss is given by

$$L_{\text{Cap}} = -\sum_{t=1}^{T} \log P_\theta(y_t \mid y_{<t}, x) \tag{3}$$

where

- T is the total number of tokens in the target sequence,

- $y_t$ is the token we are trying to predict at index $t$,

- $Y_{<t}$ is all tokens in the sequence previous to index $t$,

- $x$ is the input image we are generating the caption for, and

- $P_\theta$ is the probability of the model with parameters $\theta$ predicting $y_t$.

In practice, the cross-entropy loss function (softmax operation combined with negative log likelihood loss) is applied to the logits from the model. Three tokens were ignored in the causal modeling, which were the BOS, PAD, and CLS tokens. Accounting for this yields the final generative captioning loss function below.

$$\mathcal{L}_{\text{cap}} = -\sum_{t=1}^{T} y_t \cdot \log \left( \frac{\exp(\hat{y}_t)}{\sum_{t=1}^{T} \exp(\hat{y}_t)} \right) \cdot 1_{\{y_t \neq \text{ignore\_index}\}} \tag{4}$$

- T is the total number of tokens in the target sequence,

- $y_t$ is the element of the label vector we are trying to predict at index $t$, and

- $\hat{y}$ is the element of the logits vector we are trying to predict at index $t$.

The generative captioning evaluation metric used was perplexity, computed as the exponential of the average cross-entropy loss per token:

$$\text{perplexity} = \exp\left(\frac{\mathcal{L}_{\text{cap}}}{T}\right) \tag{5}$$

where $\mathcal{L}_{\text{cap}}$ is the summed generative captioning loss over all valid tokens, and $T$ is the number of valid (non-ignored) tokens in the sequence.

Each loss is then weighted by hyperparameters ($\lambda_{Con}$ and $\lambda_{Cap}$) and summed to produce the CoCa loss function that simultaneously minimizes both training tasks (Yu et al., 2022).

$$\mathcal{L}_{CoCa} = \lambda_{Con} * \mathcal{L}_{Con} + \lambda_{Cap} * \mathcal{L}_{Cap} \tag{6}$$

The weight of the lambdas were set to $\lambda_{Con} = 2$ and $\lambda_{Cap} = 1$ during pretraining, and $\lambda_{Con} = 1$ and $\lambda_{Cap} = 2$ during fine-tuning. This ensured pretraining focused on contrastive loss minimization and that generative captioning loss did not overfit the pretraining dataset to a large degree, allowing the model flexibility to learn the captioning of the specific fine-tuning dataset. During fine-tuning, the important function of the model is the caption it produces. Therefore, the generative captioning loss coefficient was doubled to ensure closer alignment with the objective of the model.

## 3.6 Training Techniques

The framework used for model production was PyTorch with the compute being four Tesla L40S GPUs (Paszke et al., 2019). The batch size for each GPU was 48, with 4 step gradient accumulation, making the total simulated batch size 768 samples per step ($48 \times 4 \times 4 = 768$). The optimizer used was the modern AdamW with default betas (0.9 and 0.999) and weight decay of 0.001 (Loshchilov & Hutter, 2017). The learning rate schedules used were a simple linear warm-up to maintain early stability for 2,000 steps, followed by a robust cosine annealing with warm restart scheduler (CAWR) (Loshchilov & Hutter, 2016; Liu et al., 2024). The linear warm-up was used only for pretraining. During pretraining, the maximum starting learning rate for the CAWR was 1e−4 and the minimum starting learning rate was 1e−6. During fine-tuning, the maximum and minimum CAWR learning rates were 1e−5 and 1e−7. The cycle length of the CAWR was 1 learning rate cycle per training epoch. Gradient clipping to a maximum norm of 1 was used to maintain stability.

To train until convergence on the fine-tuning datasets, a validation loss monitoring early stopper was used. The patience was set to 9 epochs on fine-tuning datasets. However, to ensure the closest possible fit to the validation partition, a soft rest mechanism was used in the early stopper. This technique reset the model to the best performing state on the validation loss if the loss had not improved over the last 3 epochs. It also lowered the maximum and minimum learning rates of the CAWR by a factor of 100 and began training again from the reset state. This allowed the model to dynamically lower the learning rate once it reached a local minimum and began alternating due to having an overly large learning rate for that geography of the parameter search space. Essentially, using this "learning rate reduction with soft reset" resulted in the model attempting to lower the validation losses at three different intervals of maximum and minimum learning rates before terminating.

## 4 Results

GRR-CoCa and Baseline CoCa were both pretrained for 21 epochs (approximately 300K steps) on the CC12M dataset. The GRR-CoCa model resulted in better scores on all evaluation metrics (CoCa loss, perplexity, and contrastive loss) on the validation partitions over the Baseline CoCa model.

Following pretraining, both Baseline CoCa and GRR-CoCa were trained to convergence on the MSCOCO, ROCO, and Flickr30K datasets. The "linear warm-up" and "learning rate reduction with soft reset" schedulers improved training time and ensured that maximal performance was gained from both GRR-CoCa and Baseline CoCa. GRR-CoCa produced better evaluation metrics across all fine-tuning datasets, following

| Metric | Baseline CoCa | GRR-CoCa | % Change |
|---|---|---|---|
| **Pretraining: CC12M Validation Set** | | | |
| CoCa Loss | 3.2864 | **3.0516** | -7.15% |
| Perplexity | 12.9976 | **12.5151** | -3.71% |
| Contrastive Loss | 0.3610 | **0.2626** | -27.25% |
| **Fine-Tuning: MSCOCO Validation Set** | | | |
| CoCa Loss | 4.6955 | **4.4090** | -6.10% |
| Perplexity | 5.4669 | **5.1523** | -5.75% |
| Contrastive Loss | 1.2971 | **1.1296** | -12.92% |
| **Fine-Tuning: ROCO Validation Set** | | | |
| CoCa Loss | 7.0609 | **6.8708** | -2.69% |
| Perplexity | 12.0258 | **11.7377** | -2.40% |
| Contrastive Loss | 2.0900 | **1.9479** | -6.80% |
| **Fine-Tuning: Flickr30K Validation Set** | | | |
| CoCa Loss | 5.4383 | **5.0111** | -7.86% |
| Perplexity | 7.9568 | **7.3686** | -7.39% |
| Contrastive Loss | 1.2908 | **1.0164** | -21.25% |

Table 2: Validation evaluation metrics of Baseline CoCa and GRR-CoCa on the pretraining (CC12M) and fine-tuning (MSCOCO, ROCO, and Flickr30K) datasets. GRR-CoCa performed better across all evaluation metrics on all datasets.

the trends of the pretraining results. The evaluation metrics and improvements relative to the baseline are shown in Table 2.

On CC12M, GRR-CoCa's ViT architecture showed a higher capacity to learn training data and generalize to unseen validation data, utilizing approximately the same parameter sizes, the same number of training epochs, and the same hyperparameter configurations as Baseline CoCa.

The evaluation metric that improved the most in pretraining was the contrastive loss. Note that the generative captioning loss was weighted half that of the contrastive loss due to the choice of lambdas ($\lambda_{\text{Cap}} = 1$, $\lambda_{\text{Con}} = 2$). GRR-CoCa made major improvements in the model's ability to achieve lower contrastive losses with almost no increase in model size. This is likely due to the fact that the contrastive task is easier for the network to model compared to capturing the complex interdependencies between natural language and images, as is done in the generative task. The perplexity also improved, but less drastically. Therefore, the modifications to the ViT improved both contrastive and generative captioning tasks in this CoCa model. This trend held true for the fine-tuning datasets where the lambda weights were opposite ($\lambda_{\text{Cap}} = 2$, $\lambda_{\text{Con}} = 1$). GRR-CoCa was shown to outperform Baseline CoCa on all evaluation metrics.

## 5 Discussion

The resulting improvements in both contrastive and generative captioning tasks for GRR-CoCa indicate that the incorporation of GEGLUs, RoPe, and RMSNorms to the ViT produces more feature-rich latent representations of images than the original ViT architecture used in Baseline CoCa. These results demonstrate that GRR-CoCa learns faster and more in-depth than Baseline CoCa.

We show that RoPe tends to provide better positional encoding than absolute positional encoding across diverse tasks and modalities. Our specific RoPe implementation injects positional encoding in each ViT block. This allows the model to preserve positional information of the patch embeddings into deeper layers by reinjecting positional information in each block. A likely explanation is that absolute positional encoding information is attenuated in deeper transformer layers as the embeddings undergo subsequent blocks, whereas RoPe maintains the ability to preserve the information, contributing to its effectiveness.

The improvements sourced from the GEGLUs are rather ambiguous, but mainly attributed to the information bottleneck they create. GEGLUs gate certain information from passing through them, effectively magnifying or minimizing said information in each embedding. We theorize this attribute must allow for more nuanced manipulation of each embedding depending on the result from the attention layer. The information filter likely facilitates more generalizable information from each embedding to be preserved and filters out noise in the embeddings, resulting in faster learning and better generalization.

RMSNorm's simplification of LayerNorms simply eliminates the recentering parameters. The increase in performance suggests that the main parameter of importance in this layer is the gamma scaling parameter from the LayerNorm instead of the beta parameter. The removal of the beta parameter causes less noise to be modeled in the training data, improving overall generalization.

## 6 Conclusions

### 6.1 Impact

Multimodal models have demonstrated broad utility not only in paired vision-language tasks but also as components in single-modality pipelines, where their pretrained visual encoders or textual decoders can be fine-tuned for specialized applications (Yu et al., 2022; Radford et al., 2021). In this work, we demonstrated that architectural enhancements to the ViT, specifically, GEGLUs, RoPe, and RMSNorms, substantially increase the robustness and downstream performance without increasing model size. Relative to Baseline CoCa, our GRR-CoCa achieved average improvements to CoCa losses, perplexities, and contrastive losses of 5.95%, 4.81%, and 17.05%, respectively, over one pretraining and three diverse fine-tuning datasets. These modifications yielded consistent gains on both contrastive objectives and generative captioning tasks, and deepened the model's learned representations. By demonstrating that GEGLUs, RoPe, and RMSNorm can be seamlessly integrated into any ViT-based visual encoder and quantifying their impact empirically, we provide practical guidance for designing next-generation foundational models. Incorporating these techniques leads to faster training, lower loss, smaller high-performance models, and improved accuracy across a wide range of downstream applications.

### 6.2 Future Work

There are several promising directions for future work that can build upon the results and insights of this study. First, future research should investigate the scaling of both model size and dataset complexity with the modified textual decoders and visual encoder modifications applied. GRR-CoCa could then be retrained on the JFT dataset and benchmarked on the same tasks presented in the original CoCa paper (Yu et al., 2022). This would provide a comprehensive review of how these modifications affect the performance of multimodal models. Selecting a larger foundational dataset like LAION-5B or JFT 3B to pair with a larger CoCa model (2B parameters) would also add more support to our current findings (Schuhmann et al., 2022; Yu et al., 2022).

Finally, beyond simply scaling, future work should also explore the adaptability of these modifications to other vision tasks such as visual question answering, visual entailment, or cross-modal retrieval. This would help determine whether the improvements observed in image-captioning generalize across tasks that also require a nuanced understanding of image and text relationships.

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
