# OpenReview forum: "GRR-CoCa: Leveraging LLM Mechanisms in Multimodal Model Architectures"
_TMLR — Withdrawn by Authors_

### Review · Reviewer_kcD9 · 2025-11-11

**Summary Of Contributions:**

This paper proposes GRR-CoCa, a CoCa-based multimodal architecture that integrates three LLM-inspired components across both modalities: GEGLU feed forward layers, RMSNorm, and RoPE.  It compared GRR-CoCA with a baseline CoCA, where the textual decoders are modified. Experiments show that GRR-CoCA is consistently and significantly better than baseline CoCA, showing that transplanting modern LLM mechanisms into ViT improves visual representations and downstream performance.

**Audience:**

Yes

**Audience Explanation:**

While this paper provides some interesting results, I have the following comments on the results and impact of the work:
1. The paper’s discussion looks limited to examining whether updating several components of CoCa will improve its performance. It would be better to highlight how the updates can be more broadly extended to other models.
2. The work mainly focuses on the improvements to CoCa, which is somewhat narrow. Other general multi-modal language models such as Qwen-VL, gpt-4o, Gemini, and latest image captioner like Blip families [1,2] show better performance on the open-end image caption tasks. Can the authors comment whether their results can be further extended to other multi-modal language models?
3. The literature review can more broadly discuss the recent literature on how modern LLM mechanisms can improve ViT. For example, Rope and RMSNorm have been used in the visual encoder of Qwen2.5-VL [3] and DINOv3 [4]. Their activation function in FFN block has been replaced with SwiGLU as well. These can be better discussed in the draft.
4. The evaluation metrics only include loss value and perplexity. The loss value may not properly reflect the actual model performance. Also, perplexity mostly reflects how the model agrees on the ground truth labels. To examine the performance of the model on real applications,  metrics including BLEU, CIDEr, and human/LLM evaluation can be utilized.
5. Ablation studies regarding the contribution of each individual modification are not provided in the numerical discussion.

[1] Li, Junnan, et al. "Blip-2: Bootstrapping language-image pre-training with frozen image encoders and large language models." ICML 2023.

[2] Xue, Le, et al. "Blip-3: A family of open large multimodal models." ICCV 2025.

[3] Bai, Shuai, et al. "Qwen2. 5-vl technical report." arXiv preprint arXiv:2502.13923 (2025).

[4] Siméoni, Oriane, et al. "Dinov3." arXiv preprint arXiv:2508.10104 (2025).

**Broader Impact Concerns:**

There is no ethical concerns.

**Claims And Evidence:**

Yes

**Claims Explanation:**

The conclusion is drawn through rigorous experiments. The implementation details, including most hyper-parameters, are clearly stated. The experiments are conducted on multiple datasets, across in-domain and out-domain datasets, which are comprehensive.

**Requested Changes:**

Following my previous comments, I request authors to make changes including numerical discussion about other ViT models and more architecture modifications, metrics such as BLEU, CIDEr for the evaluation, and conducting ablation studies regarding the contribution of individual modifications.

---

> ### Author Response · Authors · 2025-11-27
> **Author Response to Review kcD9**
>
> Thank you for your detailed review and recommendations. After careful consideration, we’ve decided to withdraw the paper for the time being, allowing us to continue developing and strengthening the work.

---

### Review · Reviewer_ZuHN · 2025-11-12

**Summary Of Contributions:**

This paper introduces GRR-CoCa, an improved version of the Contrastive Captioner (CoCa) model that integrates three architectural elements from large language models: Gaussian Error Gated Linear Units (GEGLUs), Root Mean Squared Normalization (RMSNorm), and Rotary Positional Embeddings (RoPE). The authors evaluate GRR-CoCa against a baseline CoCa model across pretraining and fine-tuning datasets. Results show consistent improvements across all metrics, with particularly large gains in contrastive loss (up to 27% during pretraining). These improvements are achieved with only a 0.17% increase in parameters.

Strengths:
Clear incorporation of proven LLM techniques into a multimodal context.
Strong empirical evidence, showing improvements are consistent across multiple datasets and evaluation metrics.
Fair baseline comparisons with matched hyperparameters and parameter counts.
Reproducible experimental setup with standard datasets and losses.

Weaknesses:
Limited novelty, as the architectural modifications are borrowed directly from LLM research and applied with no further changes.
No ablation isolating the contribution of each of the three modifications (GEGLU, RMSNorm, RoPE).

**Audience:**

Yes

**Audience Explanation:**

While the architectural novelty is modest, the paper provides clear empirical evidence that design choices from LLM research can enhance multimodal models without increasing parameter count. This insight is practical and relevant, given ongoing efforts to unify vision and language architectures. Researchers interested in efficient multimodal systems could build directly on these findings.

**Broader Impact Concerns:**

None.

**Claims And Evidence:**

Yes

**Claims Explanation:**

The claims that the proposed architectural modifications improve performance are supported by clear evidence. The paper provides detailed methodology and controlled comparisons against a baseline, with quantitative results showing consistent improvements across multiple datasets. The reported improvements in contrastive loss and perplexity are significant and consistent, supporting the claim that the architectural updates are beneficial.

However, the results are less convincing because of the limited ablation studies; the three additions together improve model performance, but they are not individually validated through controlled experiments isolating each component’s contribution.

**Requested Changes:**

[Critical] Evaluate the individual and combined impact of GEGLU, RMSNorm, and RoPE through ablations. This would strengthen causal claims and clarify which modification drives the observed improvements.

[Critical] Include variance or confidence intervals across multiple runs to ensure the reported percentage improvements are statistically meaningful.

[Optional] Include sample captions or visualizations to demonstrate whether improvements are reflected qualitatively, not just loss metrics.

[Optional] Provide mechanistic analysis (e.g., probing attention maps or embedding spaces) to support the theoretical explanations in the discussion section.

---

> ### Author Response · Authors · 2025-11-27
> **Author Response to Review ZuHN**
>
> Thank you for your time and thoughtful response. After careful consideration, we’ve decided to withdraw the paper for the time being, allowing us to continue developing and strengthening the work.

---

### Review · Reviewer_HfzR · 2025-11-13

**Summary Of Contributions:**

The paper revisits the CoCa (Contrastive Captioner) vision–language model and proposes a variant termed GRR-CoCa, which replaces several standard transformer components, i.e. LayerNorm, GELU feed-forwards, and absolute positional embeddings, with RMSNorm, GEGLU feed-forward blocks, and Rotary Positional Embeddings (RoPE), respectively. The goal is to bring “LLM-era” architectural choices into both the text decoder and the ViT image encoder. Under a fixed CoCa training recipe on CC12M and several fine-tuning datasets, GRR-CoCa shows lower contrastive and captioning losses compared to a baseline CoCa that uses only decoder-side upgrades, suggesting that modern transformer blocks can improve representation quality on the vision side as well.

Strengths:
•	The intervention is clean and well-controlled: replacing core transformer components while keeping the overall CoCa architecture, data, and objectives constant.
•	The experiments show consistent reductions in training and fine-tuning losses with no increase in model size.
•	The paper provides a practical message for practitioners: LLM-style components (RMSNorm, GEGLU, RoPE) can be dropped into ViT-based VLMs and yield immediate training benefits.
•	The write-up is clear, reproducible, and the design choices are motivated by recent trends in LLMs and ViTs.

Weaknesses:
•	The evaluation is narrow: only loss metrics are reported, with no standard retrieval, captioning, or zero-shot classification benchmarks, making it difficult to gauge the practical impact of the architectural changes.
•	The analysis attributes improvements largely to “positional preservation,” but this explanation is theoretically incomplete given recent RoPE analyses that emphasize relative attention geometry, frequency structure, and inductive biases rather than simple reinjection of positional signals.
•	There is no comparison to modern “training-trick” baselines (e.g., SigLIP loss, improved sampling, curated data), which typically produce larger gains than block-level architectural changes.
•	Ablations do not isolate the contributions of RMSNorm vs GEGLU vs RoPE in the ViT, making the mechanism underlying the improvements unclear.
•	The scale of training (CC12M) and recipe (vanilla CoCa) are slightly below current standards, limiting generalizability.

**Additional Comments:**

The paper presents a clear and practical contribution: showing that modern LLM-style components (RMSNorm, GEGLU, RoPE) can provide measurable improvements when integrated into CoCa-style multimodal models. The work is easy to read and reproducible.

That said, several claims, e.g. about why RoPE helps, would benefit from a more cautious framing or a brief acknowledgement of recent, more nuanced analyses of positional encoding behavior. Likewise, the paper would feel better grounded if positioned against the current landscape of VLM improvements, where many gains come from objective design (e.g., SigLIP) or data curation rather than architectural swaps.

Overall, the empirical core is solid for the scale studied, and with a broader evaluation and slightly refined explanations, the paper could serve as a useful reference for practitioners updating legacy CoCa/CLIP pipelines.

**Audience:**

Yes

**Audience Explanation:**

At least some segments of TMLR’s audience would be interested in the findings of this paper, though the interest is likely to be practical and engineering-oriented, rather than conceptual or theoretical.

Specifically:
•	Researchers working on vision–language models (VLMs), CLIP/CoCa variants, or multimodal transformers would find the results relevant because the paper provides clean empirical evidence that several LLM-era architectural components (RMSNorm, GEGLU, RoPE) offer drop-in improvements for ViT-based encoders. This is operationally useful, particularly for labs maintaining legacy CoCa/CLIP pipelines on modest compute budgets.
•	Practitioners focused on efficient model tuning or architecture modernization may appreciate the demonstration that architectural upgrades alone, without changing objectives or data, can give measurable improvements, especially in resource-limited or medium-scale training regimes.
•	Researchers studying the transfer of transformer design principles across modalities (language → vision → multimodal) may be interested in the empirical confirmation that LLM design innovations remain beneficial when transplanted into visual backbones.

That said, the audience interested in deeper theoretical insights, new objectives, or strong empirical benchmarks may find the contribution modest, since the work does not introduce new conceptual ideas, does not analyze mechanisms in depth, and evaluates only on a narrow set of loss metrics at small scale.

In summary:
The contribution is incremental but practically relevant, and fits the subset of TMLR readers who value empirical transformer design insights and reproducible engineering improvements in multimodal architectures.

**Broader Impact Concerns:**

The paper fine-tunes and extends CoCa-style vision–language models, which can inherit the same ethical risks associated with large-scale image–text pretraining. While this work focuses on architectural modifications and does not aim to improve or alter the model’s semantic behaviour, the authors may explicitly acknowledge that improved training efficiency and representational quality can indirectly scale up downstream deployment and, consequently, scale associated risks. It may be appropriate to add a short Broader Impact Statement noting dataset-related biases, downstream misuse risks, and the importance of responsible evaluation and deployment.

**Claims And Evidence:**

Yes

**Claims Explanation:**

Overall, the claims are partially supported by the evidence, but not to the level I would call fully convincing, mainly due to the narrowness of the evaluation and some overreach in the interpretation.

On the positive side, the core empirical claim, that is swapping LayerNorm→RMSNorm, GELU→GEGLU, and absolute PE→RoPE in the CoCa-style architecture reduces contrastive and captioning losses at fixed data and parameter count , is supported. The comparison to a baseline CoCa variant is reasonably controlled, and the loss curves clearly show consistent improvements across pre-training and fine-tuning runs. For the limited setting considered (CC12M + specific caption datasets, fixed objectives), the evidence that “GRR blocks help under this recipe” is solid.

However, several broader claims are weakly supported:
•	The paper implicitly suggests that RoPE “better preserves positional information across layers” and that this explains its superiority over absolute PEs. This is stated as an explanation rather than a hypothesis, but is backed only by aggregate loss metrics, without any targeted analysis of positional information (e.g., probing, attention-pattern analysis). Given recent, more nuanced RoPE analyses [1-3], this reads as an oversimplified and under-evidenced story.
•	Claims of improved performance “across diverse tasks and modalities” are not justified by the experiments, which are limited to a small set of image–text benchmarks, all trained under one CoCa-style objective. There are no zero-shot evaluations [4], no cross-domain/cross-dataset generalisation tests [5-6], and no analysis on tasks that stress positional structure.
•	The paper positions GRR-CoCa as a generally better architectural choice for VLMs, yet all comparisons are to a single baseline trained with an older recipe (vanilla CoCa loss, CC12M-scale data). There is no comparison to stronger “training-trick” baselines (e.g., SigLIP loss, improved sampling, curated data), which are known to yield larger gains than block-level changes. Without those, it is unclear whether the observed improvements would remain meaningful in a modern setup.
•	The ablations do not disentangle the contributions of RMSNorm, GEGLU, and RoPE on the vision side, so the causal story that LLM-style blocks in the ViT are what matter is only indirectly supported.

In summary, the narrow empirical core claim is well supported by the reported experiments, but the stronger, more general claims about why RoPE helps, how positional information is preserved, and how broadly applicable these architectural swaps are are not fully substantiated by the current evidence and would require richer metrics, stronger baselines, and more diagnostic analysis.

Refs:
[1] Barbero, Federico, et al. "Round and round we go! what makes rotary positional encodings useful?." arXiv preprint arXiv:2410.06205 (2024).
[2] Heo, Byeongho, et al. "Rotary position embedding for vision transformer." European Conference on Computer Vision. Cham: Springer Nature Switzerland, 2024.
[3] Kumar, et al., "Positional Embeddings in Transformer Models: Evolution from Text to Vision Domains", ICLR Blogposts, 2025.
[4] Cherti, Mehdi, et al. "Reproducible scaling laws for contrastive language-image learning." Proceedings of the IEEE/CVF conference on computer vision and pattern recognition. 2023.
[5] Zhai, Xiaohua, et al. "Sigmoid loss for language image pre-training." Proceedings of the IEEE/CVF international conference on computer vision. 2023.
[6] Tschannen, Michael, et al. "Siglip 2: Multilingual vision-language encoders with improved semantic understanding, localization, and dense features." arXiv preprint arXiv:2502.14786 (2025).

**Requested Changes:**

Critical changes below focus on broadening evaluation, strengthening baselines, providing proper ablations, correcting the mechanistic explanation of RoPE, and demonstrating generality.
Non-critical changes below would improve clarity, theoretical grounding, and reproducibility.


Critical changes

1. Add standard downstream evaluation metrics (not only losses).
The current evidence relies entirely on contrastive loss, captioning loss, and perplexity. To evaluate whether GRR-CoCa actually improves model utility, the paper should report at least some of:
•	Image–text retrieval: Recall@1/5/10 on COCO/Flickr30K
•	Zero-shot classification: ImageNet (and ideally ImageNet-V2/A/R)
•	Captioning metrics: CIDEr, SPICE, BLEU
These are standard in the VLM literature and are necessary for convincing conclusions.

2. Include comparisons to modern objective-level baselines (e.g., SigLIP).
Architectural block swaps must be evaluated against stronger training recipes that currently dominate improvements in VLMs such as :
•	Sigmoid contrastive loss (SigLIP)
•	Stronger negative-sampling strategies or memory banks
•	Temperature tuning or large-batch contrastive settings
Without these baselines, it is unclear whether GRR-CoCa improves over realistic state-of-the-art setups.
This comparison is important to interpret the value of the claimed improvements.

3. Provide ablations isolating the contributions of RoPE, RMSNorm, and GEGLU in the ViT encoder.
The current ablation compares:
•	baseline CoCa,
•	GRR upgrades in decoder only,
•	GRR upgrades in decoder + ViT.
But to understand why the model improves, each block change should be isolated:
•	RMSNorm-only
•	GEGLU-only
•	RoPE-only
This is critical for validating the mechanistic story and the main architectural claim.



Non-critical but recommended improvements (strengthen the work)

4. Broaden and strengthen the experimental setup.
Training only on CC12M with older CoCa objectives limits generalizability.
At least one of:
•	A larger-scale experiment (e.g., LAION-40M subset),
•	Or a transfer experiment to a stronger caption dataset (e.g., NoCaps, Conceptual Captions)
is needed to show the effect is not dataset-specific.

5. Clarify and temper the explanatory claim about RoPE (“better positional retention”).
The explanation provided (“RoPE reinjects positional info each block, preventing attenuation”) is not supported by analysis and seems too simplistic given recent research.
The authors should:
•	either substantially refine the explanation (relative geometry, frequency structure, positional heads), or
•	frame it explicitly as a hypothesis instead of a causal explanation.

7. Discuss the interaction with patch activation statistics.
Recent mechanistic analyses of transformer instabilities, including attention sinks [7] massive activations and value drift [8-9], and RoPE-induced positional discontinuities [10], show that normalization, gating, and positional encodings interact strongly with the activation geometry of the underlying representations. Since ViT patch embeddings differ from LLM token embeddings, the effectiveness of RMSNorm, GEGLU, and RoPE may depend on these modality-specific statistics. A brief discussion of these differences would help contextualize why LLM-style components transfer (or do not transfer) to ViT encoders and better situate the work within the emerging literature on transformer activation pathologies and positional-geometry behavior.

Refs:
[7] Xiao, Guangxuan, et al. "Efficient streaming language models with attention sinks." arXiv preprint arXiv:2309.17453 (2023).
[8] Guo, Zhiyu, Hidetaka Kamigaito, and Taro Watanabe. "Attention score is not all you need for token importance indicator in kv cache reduction: Value also matters." arXiv preprint arXiv:2406.12335 (2024).
[9] Sun, Mingjie, et al. "Massive activations in large language models." arXiv preprint arXiv:2402.17762 (2024).
[10] Wei, Xilin, et al. "VideoRoPE: What Makes for Good Video Rotary Position Embedding?." arXiv preprint arXiv:2502.05173(2025).

---

> ### Author Response · Authors · 2025-11-27
> **Author Response to Review HfzR**
>
> Thank you for your time and comprehensive review. After careful consideration, we’ve decided to withdraw the paper for the time being, allowing us to continue developing and strengthening the work.

---

### Review · Reviewer_3hn9 · 2025-11-16

**Summary Of Contributions:**

This paper introduces **GRR-CoCa** , a multimodal architecture that integrates modern LLM architectural advances— Gaussian Error Gated Linear Units (GEGLU), Root Mean Square Normalization (RMSNorm), and Rotary Positional Embeddings (RoPE)—into the Vision Transformer (ViT) backbone of the foundational CoCa model. The core idea presented is that while multimodal models like CoCa achieve strong performance, their foundational architectures often lag behind the sophistication of contemporary LLMs, and thus provides reason to integrate the architectural advances that led to their improvements.

**Key Strengths :**
- applied methods achieve consistent gains across losses
- appropriate testing on a wide range of comprehensive datasets
- clearly outlined experimental specifications (hyper-parameters, compute requirements)

**Key weaknesses :**
-  ablations separately across the three introduced improvements are absent
- qualitative examples on differential edge cases showing practical improvements should be provided
- deeper mechanistic reasoning about the mechanisms these modifications enable in the image(multimodal) space in contrast to the language space is absent

**Additional Comments:**

N/A

**Audience:**

Yes

**Audience Explanation:**

Readers interested in design improvements for VLMs at large, and multimodal architectures would find promise in the gains presented by this paper.

**Claims And Evidence:**

Yes

**Claims Explanation:**

The presented experiments are thorough with evaluations across comprehensive datasets, considering the scope that the paper presents. The claims of improvements induced by these architectural modifications are backed by the gains across losses and datasets in the experimental results. The explicit details about hyperparamters, training regime and compute requirements are welcomed.

Despite the strong empirical results datasets, I would still suggest that the evidence is not entirely complete in its presentation. Their are a number of gaps that undermine the strength of some of their claims :

**Lack of ablation studies:** The paper introduces three architectural modifications (GEGLUs, RMSNorm, RoPE) simultaneously to the ViT encoder without testing individual components. Since this a paper focused more on design improvements, the claim that these modifications improve performance cannot be differentiated from the possibility that only one or two components drive the improvements, or that interaction effects between components are responsible.

**Mechanistic Reasoning could be stronger:** While positioning itself primarily as an improvement to CoCa, the paper does claim a more general impact of their architectural incorporations to be missing from current vision/multimodal architectures. While these inclusions do bring about considerable improvements, the manuscript offers little/high level explanations of ***why*** do they seem to work. Image/Multimodal spaces are fundamentally different from language spaces where these were first introduced and does not necessarily have many transferable properties (eg.: 2D layouts vs sequential information). Explanations on how these enable more competent mechanisms inside the model, especially in the vision space, are absent. The present explanations/hypothesis *"A likely explanation is that absolute positional encoding information is attenuated in deeper transformer layers as the embeddings undergo subsequent blocks, whereas RoPe maintains the ability to preserve the information, contributing to its effectiveness."*, *"We theorize this attribute (about GEGLU) must allow for more nuanced manipulation of each embedding"* remain rather high-level and generic.

Better supported explanations would include details about how the mechanisms enabled by these changes apply to visual information and vision tasks, and how are these mechanisms complementary to their language space counterparts. For example, the 27.25% contrastive loss improvement suggests RoPE is particularly effective for vision—does this imply 2D spatial relationships benefit more from rotary encoding than 1D sequential relationships? A comparative analytical discussion with the magnitude of RoPE improvements in comparable LLMs could be beneficial.

**Qualitative examples :** While the presented quantitative reasons are strong, their implications on actual captioning examples or testing any edge cases as compared to the vanilla CoCa method are not provided. These are usually helpful for a reader to understand how this affects the model's performance in accordance to its outputs, for eg.: did captions become more detailed? Or where they able to capture additional information previously to model oversight. To give some examples : Claims such as *"deepened the model's learned representations"* or motivations to include the changes *"to produce more feature rich representations"* could also be supported with qualitative visualizations of the features (testing clustering and separation between different model classes). The claim *"RoPe maintains the ability to preserve positional information in deeper layers"* could be supported by attention pattern visualizations, if the claim holds, GRR-CoCa should maintain more structured spatial attention in deeper layers while Baseline CoCa's attention becomes diffuse.

Overall I do find the evidence to be promising and comprehensive, but the outlined points above would make it a much more complete presentation in my view.

**Requested Changes:**

### **Major**:
- **Perform component-level ablation studies:** Add experiments isolating the individual contributions of GEGLU, RMSNorm, and RoPE within the ViT encoder
- **Deepen mechanistic analysis for vision/multimodal settings:** Provide more detailed reasoning or experiments to support how the proposed architectural modifications specifically improve visual and multimodal tasks differently from their original context in language models
- **Include qualitative evidence:** Add examples showing model outputs (e.g., side-by-side captions, hard prompts, or edge cases) and internal model dynamics (e.g., attention maps, feature visualizations) to demonstrate how GRR-CoCa qualitatively improves over Baseline CoCa.

### **Minor** :

- A comparison with vanilla CoCa (without the text decoder modification) could also be included since the current baseline provides a controlled comparison to isolate improvements in the vision space. (Could be excluded optionally provided appropriate reasoning is given)

---

> ### Author Response · Authors · 2025-11-27
> **Author Response to Review 3hn9**
>
> Thank you for your insightful comments and feedback. After careful consideration, we’ve decided to withdraw the paper for the time being, allowing us to continue developing and strengthening the work.

---

### Note · Authors · 2025-11-27

**Comment:**

After careful consideration, we’ve decided to withdraw the paper for the time being, allowing us to continue developing and strengthening the work. Thank you again to the reviewers and TMLR editors for their careful evaluations and time.

**Withdrawal Confirmation:**

I have read and agree with the venue's withdrawal policy on behalf of myself and my co-authors.